# The natural tannins oligomeric proanthocyanidins and punicalagin are potent inhibitors of infection by SARS-CoV-2

Hsiao-Fan Chen[1,2], Wei-Jan Wang[2,3], Chung-Yu Chen[2], Wei-Chao Chang[4], Po-Ren Hsueh[5], Shin-Lei Peng[6,7], Chen-Shiou Wu[1,2], Yeh Chen[3,8], Hsin-Yu Huang[1], Wan-Jou Shen[1], Shao-Chun Wang[1,2,4,9,10]*, Mien-Chie Hung[1,2,4,9,10,11]*

[1]Graduate Institute of Biomedical Sciences, China Medical University, Taichung, Taiwan; [2]Research Center for Cancer Biology, China Medical University, Taichung, Taiwan; [3]Department of Biological Science and Technology, College of Life Sciences, China Medical University, Taichung, Taiwan; [4]Center for Molecular Medicine, China Medical University Hospital, China Medical University, Taichung, Taiwan; [5]Departments of Laboratory Medicine and Internal Medicine, China Medical University Hospital, School of Medicine, China Medical University, Taichung, Taiwan; [6]Department of Biomedical Imaging and Radiological Science, China Medical University, Taichung, Taiwan; [7]Neuroscience and Brain Disease Center, China Medical University, Taichung, Taiwan; [8]Institute of New Drug Development, China Medical University, Taichung, Taiwan; [9]Cancer Biology and Precision Therapeutics Center, China Medical University, Taichung, Taiwan; [10]Department of Biotechnology, Asia University, Taichung, Taiwan; [11]Institute of Biochemistry and Molecular Biology, China Medical University, Taichung, Taiwan

*For correspondence:
scpwang@gmail.com (S-CW);
mhung@cmu.edu.tw (M-CH)

**Abstract** The Coronavirus Disease 2019 (COVID-19) pandemic continues to infect people worldwide. While the vaccinated population has been increasing, the rising breakthrough infection persists in the vaccinated population. For living with the virus, the dietary guidelines to prevent virus infection are worthy of and timely to develop further. Tannic acid has been demonstrated to be an effective inhibitor of coronavirus and is under clinical trial. Here we found that two other members of the tannins family, oligomeric proanthocyanidins (OPCs) and punicalagin, are also potent inhibitors against severe acute respiratory syndrome coronavirus 2 (SARS-CoV-2) infection with different mechanisms. OPCs and punicalagin showed inhibitory activity against omicron variants of SARS-CoV-2 infection. The water extractant of the grape seed was rich in OPCs and also exhibited the strongest inhibitory activities for viral entry of wild-type and other variants in vitro. Moreover, we evaluated the inhibitory activity of grape seed extractants (GSE) supplementation against SARS-CoV-2 viral entry in vivo and observed that serum samples from the healthy human subjects had suppressive activity against different variants of SARS-CoV-2 Vpp infection after taking GSE capsules. Our results suggest that natural tannins acted as potent inhibitors against SARS-CoV-2 infection, and GSE supplementation could serve as healthy food for infection prevention.

## Editor's evaluation

The study reports new findings that support the conclusion that natural tannins oligomeric proanthocyanidins and punicalagin are potent inhibitors of infection by SARS-CoV-2. The significant observations made by the authors could provide new strategies to prevent SARS-CoV-2 infection.

## Introduction

Since its outbreak in late 2019, the Coronavirus Disease 2019 (COVID-19) pandemic has made a significant impact threatening health care to the general livelihood of virtually everybody on earth. In the past year, deep knowledge of the biology of the virus has been unveiled, and effective vaccines against severe acute respiratory syndrome coronavirus 2 (SARS-CoV-2) have been rolling out to fight the pandemic. However, with this tremendous progress, the pandemic has been resilient and far beyond to be waning down.

The challenge of the COVID-19 pandemic has become direr with the emergence of a swarm of variants popping out in multiple countries around the globe (*Plante et al., 2021*). Some of the variants, such as the alpha variant of the UK lineage B.1.1.7 (*Zhou and Wang, 2021*), the closely related beta variant of the South Africa lineage B.1.351 (*Plante et al., 2021*), and the gamma variant of the Brazil lineage P.1 (*Coutinho et al., 2021*), have higher infection rates and are more deadly than the wild-type SARS-CoV-2 strain (*Torjesen, 2021*). The delta variant of the India lineage B.1.617.2 was first identified in India in December 2020, rapidly swept across European countries in June 2021, and surged worldwide to become the most predominant SARS CoV-2 variant with increased activity transmissibility compared to other countries' pandemic variants in 2021 (*Lopez Bernal et al., 2021*). Following the previous mutant strains, Delta infection accounts for the increasing breakthrough cases in fully vaccinated population and >99% of hospitalized cases of COVID-19, sounding the alarm that the mainstay vaccines may not be as efficient as they were expected in protection from virus infection and symptom development (*Chia et al., 2021*; *Riemersma et al., 2021*). After almost 1 y, the newest highly divergent omicron variant of the lineage B.1.1.529 was identified in South Africa, in November 2021, and quickly became the dominant variant around the world in 2022 (*Mehta et al., 2022*). The omicron variant has >50 mutations relative to the wild-type virus, with 30 nonsynonymous substitutions in the Spike protein, which interacts with human cells for cell entry and that has been the primary target for current vaccines (*Sun et al., 2022*). The robust immune evasion capability allows the omicron variant to quickly escape from existing mAbs and vaccines and increase the risk of reinfection (*Ao et al., 2022*). Thus, the dietary guidelines for preventing SARS-CoV-2 infection are of high stake.

Infection by the SARS-CoV-2 starts from the cellular entry of the virus through the interaction between the viral Spike (S) protein with its cell surface receptor ACE2 (angiotensin-converting enzyme 2) and priming of the S protein by the cellular protease TMPRSS2 (transmembrane serine protease 2) (*Hoffmann et al., 2020*). Proteolytic processing of the S protein by TMPRSS2 produces S1 and S2 fragments, which remain noncovalently associated. The S1 polypeptide interacts with the ACE2 receptor through the receptor binding domain (RBD), and the S2 polypeptide facilitates the fusion of the virus with the cell membrane (*Wang et al., 2020b*). The internalized virus translated its RNA genome into a polypeptide, followed by a highly regulated and coordinated proteolytic processing by the main protease ($M^{pro}$/$3CL^{pro}$), a chymotrypsin-like protease, and cleaves its substrate polypeptides specified by well-defined cleavage sequences (*Anand et al., 2003*; *Hilgenfeld, 2014*; *Yang et al., 2003*). Blocking the maturation process of the viral polyproteins thus abrogates viral replication in host cells (*Jin et al., 2020*). Together these viral and cellular proteins constitute promising targets for clinical treatment and prevention of SARS-CoV-2.

We previously showed that several natural products including the natural polyphenol compound tannic acid are potent inhibitors of TMPRSS2 and $M^{pro}$ (*Huang et al., 2020*; *Wang et al., 2022b*; *Wang et al., 2022a*; *Wu et al., 2022*). Tannic acid belongs to the unique family of tannins consisting of four subfamilies: hydrolysable tannins, condensed tannins, phlorotannins, and complex tannins. Hydrolysable tannins and condensed tannins are found in terrestrial plants, especially fruits and vegetables, while phlorotannins occur only in brown marine algae (*Das et al., 2020*; *Huang et al., 2018*). Compared to phlorotannins, hydrolysable tannins and condensed tannins are widely distributed in nature and are present in plant foods and some beverages (*Serrano et al., 2009*). In this report, we compared the representative derivative compounds of the tannin family, namely tannic acid,

**eLife digest** Since it first surfaced in late 2019, the COVID-19 pandemic has had a significant impact on people's lives. While several vaccines have been created, infections have not disappeared. This is largely due to new variants of the virus responsible for the disease (SARS-CoV-2) emerging, which current vaccines do not work as well against. Indeed, several reports suggest that protection from the omicron variant wanes as shortly as four to six months after vaccination. Therefore, other strategies are needed to reduce the risk of SARS-CoV-2 infections.

In 2022, researchers discovered that tannic acid blocked two proteins that SARS-CoV-2 needs to enter and replicate inside human cells. Tannic acid is part of the tannin family, which includes natural molecules found in plant-based meals and beverages. Here, Chen et al. – including some of the researchers involved in the 2022 studies – set out to find whether two other tannins found in nature (OPCs and punicalagin) could also inhibit SARS-CoV-2.

Chen et al. administered tannic acid, OPCs and punicalagin to human cells cultured in a laboratory that had been infected with SARS-CoV-2. This revealed that all three tannins suppress the activity of the same proteins required for viral entry and replication, but to varying degrees suggesting that they block SARS-CoV-2 infections via different mechanisms. The compounds were also able to inhibit different variants of the virus, including omicron, from infecting the lab-grown cells.

Further experiments revealed that water extracted from seeded grapes, which contains high levels of OPCs, could also block SARS-CoV-2 entry in the cell culture system. To test this further, Chen et al. gave 18 healthy individuals capsules containing different concentrations of grape seed extract and collected samples of their serum. The serum samples suppressed entry of different variants of SARS-CoV-2 in the cell culture system, with serums from subjects that received the higher dose having the greatest effect.

These findings suggest that naturally occurring tannins can suppress multiple variants of SARS-CoV-2 from entering and replicating in cells. Consuming supplements of grape seed extract could potentially reduce the risk of SARS-CoV-2 infections. However, further experiments, including clinical trials, are needed to test this possibility.

punicalagin, and proanthocyanidins (OPCs), of the hydrolysable and condensed tannins (*Fraga-Corral et al., 2021*). The results reveal that the tannin family is a potential resort of plant foods and beverages that may associate with activity to prevent infection of anti-SARS-CoV-2. Grape seed extract (GSE), a natural food supplement, was found to enrich OPC content, and serum from individuals who took the GSE exhibited activity to suppress infection of SASR-CoV-2.

## Methods

### Spike protein–ACE2 interaction ELISA assay

The inhibitory effects of tannic acid, OPCs, or punicalagin on the interaction between SARS-CoV-2 Spike protein and human ACE2 were detected using the SARS-CoV-2 enzyme-linked immunosorbent assay (ELISA) Kit (AllBio). Briefly, horseradish peroxidase (HRP)-conjugate-ACE2 was pre-incubated with various concentrations of tannic acid, OPCs, or punicalagin (0, 50, 100 µM) at room temperature for 30 min, followed by the addition to the ELISA plate pre-coated with SARS-CoV-2 Spike RBD at 37°C for 1 hr. Each well was aspirated and then washed with washing buffer five times. After the last wash, 90 µl TMB (3,3,5,5-tetramethylbenzidine, AllBio) substrate solution was added to each well and incubated at 37°C for 20 min. The color development was then stopped by adding a 50 µl Stop solution, and the HRP activity was measured by detecting the optical density at 450 nm. The binding rate or interaction between SARS-CoV-2 Spike RBD and human ACE2 was calculated using the following equation: Spike-ACE2 interaction (%) is the ratio of averaged OD450 values of the sample/DMSO control × 100%.

### FRET-based enzyme activity assay

The fluorescence resonance energy transfer (FRET)-based enzyme activity assay of SARS-CoV-2 Mpro was established in the previous study (*Chen et al., 2022a*; *Wang et al., 2020c*). Briefly, purified

SARS-CoV-2 Mpro was incubated with different concentrations of tannic acid, OPCs, or punicalagin (0, 50, 100 μM) in 20 mM Tris pH 8.0, 20 mM NaCl at room temperature for 30 min. Fluorescent protein substrate CFP-YFP was added to initiate the reaction. The activity assay was measured for 30 min by detecting the fluorescent signal at a wavelength of 474 nm after excitation at a wavelength of 434 nm. The reaction velocity was calculated using the data points from the first 15 min and normalized to the DMSO control.

### Cell culture

The human embryonic kidney cells 293T stably expressing recombinant human ACE2 (293T-ACE2) were maintained in DMEM containing 10% FBS, 1% P/S, and 200 μg/μl hygromycin. The NCI-H460 cell line (human non-small cell lung cancer cells) was preserved in Modified Eagle's Medium (MEM, Gibco, USA) supplemented with 10% FBS and 1% P/S. The VeroE6 cell line (African green monkey kidney cells) was grown in Dulbecco's Modified Eagle's Medium (DMEM, Gibco) supplemented with 10% fetal bovine serum (FBS, Gibco), 1× GlutaMAX (Gibco), and 1% penicillin/streptomycin (P/S, Hyclone, USA). All these cells were cultured at 37°C and 5% $CO_2$. All cell lines are checked annually for mycoplasma contamination, and human cell lines were authenticated through STR profiling.

### Cell transfection

VeroE6 and 293T cells were transfected with a pCMV3-flag-TMPRSS2 plasmid (Sino Biological) to overexpress human TMPRSS2 by using Lipofectamine 2000. One day before the plasmid transfection, cells were split into six-well plates. To prepare transfection complexes, pCMV3-flag-TMPRSS2 plasmid and Lipofectamine 2000 were mixed into OptiMEM medium and incubated at room temperature for 20 min before being added to the cells.

### Cell-based TMPRSS2 enzyme activity assay

The cell-based enzyme activity assay of TMPRSS2 was established in the previous study (*Jin et al., 2020*). 293T and 293T-TMPRSS2-expressing cells were seeded in a black, 96-well plate (20,000 cells/well). The next day, the growth medium was replaced with 80 μl of compounds or PBS alone to the wells in the indicated concentrations and incubated at room temperature for 1 hr. The 100 μM fluorogenic substrate Boc-QAR-AMC (R&D Biosystems) was added to each well and put in a 37°C incubator for another 2 hr. Fluorescence (excitation 360 nm, emission 410 nm) was measured using Synergy H1 hybrid multi-mode microplate reader (BioTek Instruments, Inc).

### Cytotoxicity assay

The cytotoxicity of tannic acid, punicalagin, or OPCs was assessed by seeding cells in 96 well plates in 100 μl media and cultured for 24 hr, followed by treatment with different doses of tannic acid, punicalagin, or OPCs in 50 μl complete medium containing for another 24 hr. The medium was removed, and viable cells were detected with MTT assay (Cyrusbioscience, Taiwan), which was read by absorbance at 570 nm by an ELISA reader. All experiments were performed independently in triplicates and repeated three times.

### Vpp assay

All SARS-CoV2-S viral pseudo-particles were purchased from the RNAi core of the Academia Sinica, Taiwan (http://rnai.genmed.sinica.edu.tw/). Briefly, cells were seeded into 96-well plates and pretreated with different doses of tannin compounds for 1 hr, then inoculated with 50 μl pseudo-virions (0.1 MOI [multiplicity of infection]). After incubation for 1 d, cell viability was confirmed by the Cell Counting Kit-8 (CCK-8) assay (Dojindo Laboratories, Japan). Each sample was mixed with 100 μl luciferase substrate (Bright-Glo Luciferase Assay System, purchased from Promega, USA), and luminescence was measured immediately using the GloMax Navigator System (Promega). Viability-normalized relative light unit was set as 100% for the control group, and the relative infection rate of the treated groups was calculated. All experiments were performed in triplicates and repeated three times independently.

### Viruses

Sputum specimens obtained from SARS-CoV-2-infected patients were maintained in a viral transport medium. The virus in the specimens was propagated in VeroE6 cells in DMEM supplemented with

2 µg/ml tosylsulfonyl phenylalanyl chloromethyl ketone-trypsin (T1426, Sigma-Aldrich, USA). Culture supernatant was harvested when cytopathic effect was observed in >70% of cells, and the virus titer was determined by plaque assay. The virus isolates used in the current study were hCoV-19/Taiwan/NTU128/2021. The virus isolation was conducted in the Biosafety Level-3 Laboratory at National Taiwan University Hospital. The study was approved by the NTUH Research Ethics Committee (202101064RINB), and the participants gave written informed consent.

## Yield reduction assay

The Vero E6 cells were seeded to the 24-well culture plate at $2 \times 10^5$ cells/well in DMEM with 10% FBS and penicillin G sodium 100 units/ml, streptomycin sulfate 100 µg/ml, and amphotericin B 250 ng/ml (antibiotic-antimycotic, Gibco) 1 d before infection. The virus (0.02 MOI) was premixed with the test compound at indicated concentrations, solvent control, or medium containing 2% FBS (E2) for 1 hr at 37°C. Then, the mixture was added to the cells for another 1 hr of incubation at 37°C. At the end of incubation, the virus inoculum was removed and the cells were washed once with PBS buffer before adding fresh E2 medium (500 µl/well) containing test compound at indicated concentrations or solvent control for 24 hr at 37°C. Finally, the cellular RNA was extracted using NucleoSpin RNA Kit (Macherey-Nagel GmbH&Co. KG, Germany) for real-time PCR analysis of SARS-CoV-2 E genes using the iTaq Universal Probes One-Step RT-PCR Kit (Bio-Rad, USA) and the QuantStudio 5 Real-Time PCR System (Applied Biosystems, USA). All of the experiments involving the SARS-CoV-2 virus were performed in the Biosafety Level-3 Laboratory of the First Core Laboratory, National Taiwan University College of Medicine.

## Preparation of grape extractant

The fresh Kyoho grapes were obtained from Xihu and Dacun Township of Changhua County and were divided into seed, peel, and flesh parts. The fresh seedless grapes were purchased from the local market in Taichung. The juice of two kinds of grapes or each part of Kyoho grapes was homogenized using a household blender (model: JJM2500, Jye Sheng Industrial Co., Taichung, Taiwan). The grape extractants were prepared by adding 20 ml water to 1 g of homogenized seed, peel, and flesh, respectively. OPCs were extracted from grape samples by shaking for 1 min and then standing for 10 min at ambient temperature. The water extractant of grapes was filtered through a 0.22 m polyvinylidene difluoride (PVDF) filter (Paul, Port Washington, NY) and subsequently analyzed by UHPLC-HRMS.

## UHPLC-HRMS analysis

The grape seed, peel, and flesh extractants were analyzed on an Ultimate 3000 UHPLC system coupled to a Q-Exactive Plus high-resolution mass spectrometer (Thermo Scientific, San Jose, CA). The separation was carried out using Waters ACQUITY UPLC C18 column (2.1 × 100 mm, 1.7 µm) as well as water containing 0.2% formic acid and acetonitrile was utilized as mobile phase A and B. The separating gradient program was started at 5% B and held for 1 min and then increased linearly to 40% B at 15 min, 80% at 16 min, and held for 5 min. The flow rate was 0.4 ml/min, and the injection volume was 5 µl. The temperatures of the column and sample were 25 and 5°C, respectively. The mass spectra were obtained using electrospray ionization in negative mode. The parameters of HRMS were as follows: spray voltage was 4.0 kV; scanning mode was full scan mode with a mass range of m/z 150–2000; and mass resolution was 70,000. The deprotonated molecules ([M−H]−) were extracted and used for the quantification of OPCs in grape water extractants. OPC standard was purchased from Sigma-Aldrich (St. Louis, MO) and used for the preparation of calibration solution in UHPLC-HRMS analysis.

## Human studies

Eighteen healthy volunteers were invited to participate. None of the participants received any pharmacological treatment during blood collection. Fruits and vitamin supplements were discontinued 24 hr prior to the collection of the first blood sample. All subjects were instructed to maintain their usual diets during the 2-day study period. All subjects provided written informed consent, and the study protocol was approved by the CMUH Research Ethics Committee (CMUH111-REC3-106). On the first day, the baseline blood sample was collected before taking control or grape GSE capsules. After baseline blood sample collection, the human subjects were randomized to one of three groups

to take the GSE capsule (200 mg or 400 mg), or the placebo and collected another blood sample after taking the capsule for 4 hr. On the second day, each volunteer took the GSE capsule or the placebo at the same time again. The third blood sample was collected after taking the capsule for 4 hr. as described above. After collection of each blood sample, serum samples were placed at room temperature for 15 min to allow proper clot formation and then centrifuged at 1000 × $g$ for 10 min in a refrigerated centrifuge. Each serum sample was diluted 1/200 for Vpp assay.

## Statistical analysis

The independent $t$-test was performed to compare the continuous variation of two groups. Extra-sum-of-squares $F$-test was applied for comparison of CC50 or EC50 curve. p-values<0.05 were considered to be significant.

## Results

### Tannic acid, punicalagin, and OPCs had different abilities for blocking Spike–ACE2 interaction, Mpro, or TMPRSS2 inhibition

The potential of the natural tannins, including tannic acid, punicalagin, and OPCs, in inhibiting SARS-CoV-2 infection was assessed starting by measuring their effects at different doses (0, 50, and 100 μg/ml) in the recruitment of the Spike protein to ACE2, the critical early step for viral entry. The binding of the two proteins was measured using an ELISA-based analysis (*Abe et al., 2020*; *Amanat et al., 2020*). The results showed that punicalagin had the highest inhibitory activity against the Spike protein–ACE2 interaction compared to tannic acid and OPCs (*Figure 1A*). This result suggested that the tannins, significantly punicalagin, could block viral infection by interrupting Spike–ACE2 interaction. However, it does not exclude the involvement of other mechanisms. To further test the mode of action of the tannins, the enzymatic activities of M^pro were assessed using a FRET-based assay method developed in prior studies (*Chen et al., 2021*; *Chen et al., 2022b*). OPCs significantly inhibited M^pro as potent as tannic acid, a potent M^pro inhibitor as previously reported (*Wang et al., 2020d*; *Figure 1B*). Interestingly, punicalagin exhibited even stronger inhibitory activity on M^pro than tannic acid and OPCs. Tannic acid had been reported as a dual inhibitor to M^pro and TMPRSS2. Therefore, we perform a FRET-based assay to measure the TMPRSS2 proteinase activity in vivo. In contrast, the result showed that punicalagin had only moderate inhibitory activity on TMPRSS2 compared to tannic acid and OPCs (*Figure 1C*). Thus, all three tannins are able to suppress M^pro and TMPRSS2 activities with punicalagin exhibiting the strongest suppressive activity of M^pro, but the least potency to suppress TMPRSS2 activity.

### Tannic acid, OPCs, and punicalagin could inhibit the SARS-CoV-2 pseudo-viral entry through different mechanisms

These in vitro results suggested an anti-SARS-CoV-2 potential of the tannins in vivo. For evaluating the efficacy of viral infection, an assay was developed to measure cell entry by the pseudoviral particle (Vpp) of SARS-Co-V-2, entailing the mammalian cell lines 293T-ACE2 (human embryonic kidney cells expressing ACE2) and NCI-H460 (human non-small cell lung cancer cells). A cytotoxicity assay was performed to clarify the nontoxic concentration of tannic acid, OPCs, and punicalagin in these two cell lines. The results showed that treatments with OPCs and punicalagin resulted in better viability than tannic acid in both cell lines (*Figure 2A–F*). Nevertheless, three tannin treatments in 293T-ACE2 cells showed lower toxicity than NCI-H460 (*Figure 2A–F*). According to cell viability results, most cells survived the three tannins at doses lower than 50 μg/ml in these three cell lines. Therefore, the doses of relatively low toxicity (0, 2, 10, and 50 μg/ml) were chosen to conduct the viral entry assays. As shown in *Figure 2*, treatments with tannic acid, OPCs, and punicalagin resulted in a dose-dependent inhibition of the Vpp infection in 293T-ACE2 and NCI-H460 cells. *Figure 2G* summarizes the 50% cytotoxic concentration (CC50), 50% effective concentration (EC50), and selectivity index (SI) from the experiments in *Figure 2A–F*. Compared to tannic acid, the EC50 of OPCs was slightly increased in both cell lines, and punicalagin was slightly increased in NCI-H460 cells. In the treatment of 293T-ACE2 cells, punicalagin exhibited the highest SI index because the CC50 of punicalagin in 293T-ACE2 cells was the highest than in the other cell lines. These results suggested that tannic acid, OPCs, and punicalagin could inhibit the viral pseudo-particles (Vpps) infection.

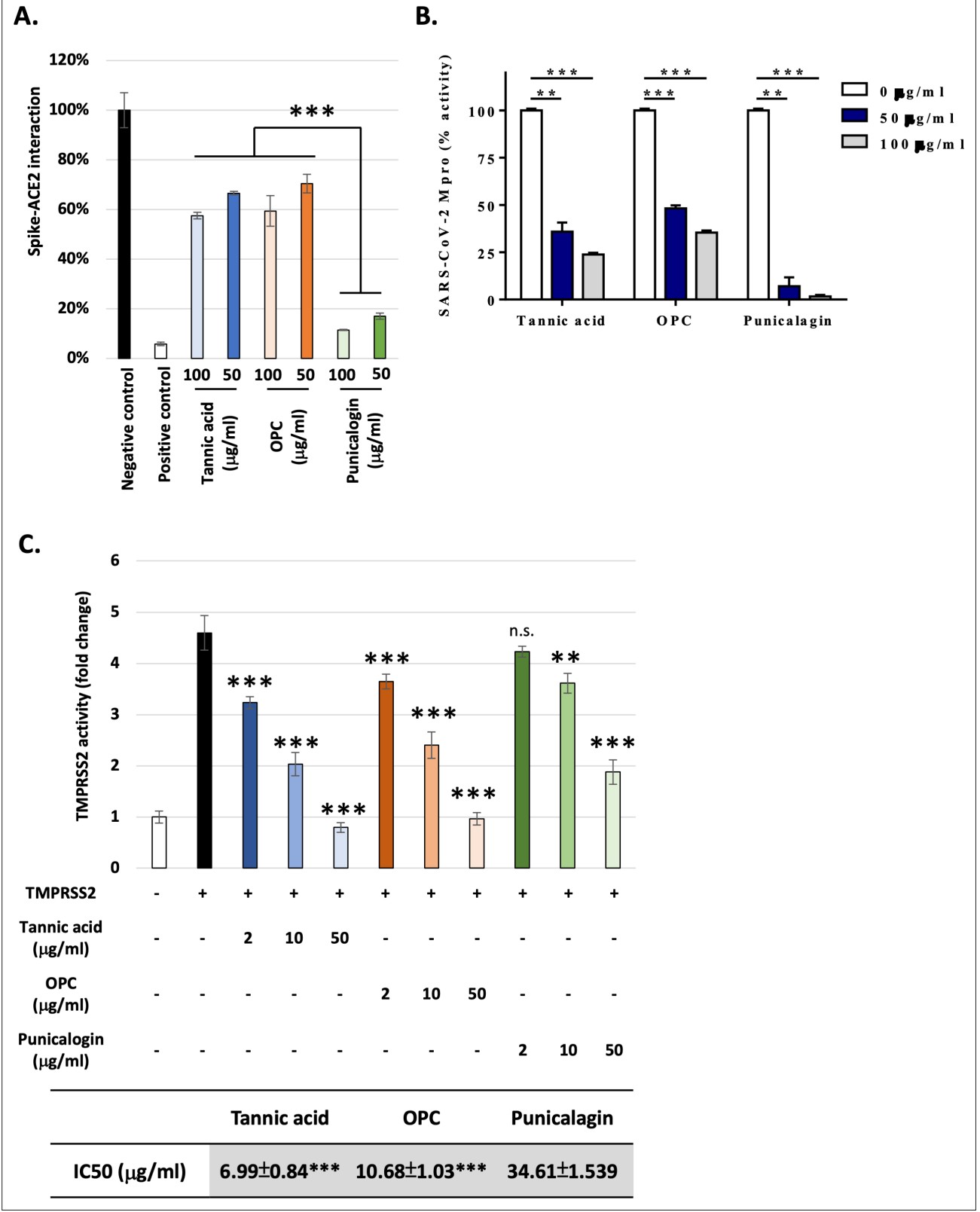

**Figure 1.** Punicalagin efficiently blocked ACE2–Spike protein interaction and repress the main protease activity, while tannic acid and oligomeric proanthocyanidins (OPC) had higher inhibition activity against TMPRSS2. (**A**) The percentage of Spike–ACE2 interaction from ELISA-based assay is shown with the indicated concentration of tannic acid, OPCs, or punicalagin. All data are shown as mean ± SD (n = 3). The p-values are indicated by asterisks, ***p≤0.001.(**B**) The main protease enzymatic activity was measured using a FRET-based assay with the indicated concentration of tannic

*Figure 1 continued on next page*

*Figure 1 continued*

acid, OPCs, or punicalagin. All data are shown as mean ± SD (n = 3). The p-values are indicated by asterisks, **p≤0.01; ***p≤0.001. (**C**) The TMPRSS2 enzymatic activity in vivo was measured using a FRET-based assay with an increasing amount of tannic acid, OPCs, or punicalagin. All data are shown as mean ± SD (n = 3). The p-values are indicated by asterisks compared to TMPRSS22 alone group (black column). Extra-sum-of-squares *F*-test was performed to evaluate differences of IC50 of tannic acid or OPC compared to punicalagion. **p≤ 0.01; **p≤ 0.01; ***p≤0.001; n.s., no statistical significance.

The online version of this article includes the following source data for figure 1:

**Source data 1.** The table displays the inhibitory concentrations (IC50) of tannic acid, OPCs, and punicalagin for TMPRSS2 activity.

To further address the effects of tannins on the inhibition of TMPRSS2 activity, human TMPRSS2 or a control vector were ectopically expressed in the VeroE6 cell lines (African green monkey kidney epithelial). First, the results showed that treatments with tannic acid, OPCs, and punicalagin resulted in similar viability (CC50) in the VeroE6 cell line (*Figure 3A and E*) compared to 293T-ACE2 and NCI-H460 cell lines (*Figure 2G*). For both tannic acid (*Figure 3B*) and OPCs (*Figure 3C*) but not punicalagin (*Figure 3D*), cells expressing TMPRSS2 were more sensitive to the treatment, possibly due to the improved assay sensitivity in the presence of TMPRSS2. The EC50 and SI values of tannic acid and OPCs in the VeroE6 cells expressing TMPRSS2 were significantly better than that in VeroE6 control cells (*Figure 3E*). This result supports that tannic acid and OPC had higher potency to inhibit TMPRSS2 enzyme activity in *Figure 1C*. In conclusion, tannin derivatives may be potential inhibitors for SARS-CoV2 cell entry by blocking Spike protein–ACE2 interaction and TMPRSS2 enzymatic inhibition.

## Tannic acid and OPCs could maintain inhibitory activities against different variants of the SARS-CoV-2 pseudo viral entry

The inhibitory activity of the tannins to viral infection was not limited to the Vpp of the wild-type SARS-CoV-2 strain. Vpp derived from the alpha, beta, gamma, delta, or omicron variants were tested for infection efficiency in the 293T-ACE2 and NCI-H460 cell lines. Treatment with tannic acid, OPCs, and punicalagin resulted in significant suppression of infection by the Vpp derived from SARS-CoV-2 wild type (*Figure 4A*), the alpha variant (*Figure 4B*), or the gamma variant (*Figure 4D*). It should also be noted that punicalagin was less potency in suppressing entry of Vpp derived from the beta variant (*Figure 4C*), the delta variant (*Figure 4E*), and the omicron variant (*Figure 4F*) in both 293T-ACE2 and NCI-H460 cell lines than the other two compounds. The in vitro data shown in *Figure 1* indicated that punicalagin had higher inhibitory activity to block Spike protein–ACE2 interaction but less effect on TMPRSS2 enzymatic activity. These results implied that punicalagin might lose its inhibitory activity for viral entry of variants such as beta, delta, and omicron strains likely because of different mutations on the Spike protein of SARS-CoV-2 variants. In contrast, tannic acid and OPCs had a more substantial effect on suppressing TMPRSS2 enzyme activity and then could maintain their potency for inhibiting different SARS-CoV-2 variants' viral entry.

## All three tannin compounds could block SARS-CoV-2 omicron variants infection by inhibiting viral entry or replication

The Vpp infection assay only can evaluate the efficacy of tannic acid, punicalagin, and OPCs for the viral entry. Next, we performed a virus yield reduction assay to test the inhibitory activity of these three tannins compounds by using a live virus. As shown in *Figure 5A*, different concentrations of tannic acid, punicalagin, or OPCs were treated with the lineage omicron virus for 1 hr. Then VeroE6 cells were infected by the mixtures containing the virus and tannins for another hour. Cells were continuously treated with the same concentration of tannic acid, punicalagin, or OPCs for 1 d after the 1 hr infection. Cellular RNA was collected to further quantify virus yield by measuring viral RNA of nucleoprotein (N) using quantitative real-time RT-PCR (qRT-PCR). The results indicated that all three tannin compounds significantly repressed the yield of viral RNA (*Figure 5B–D*). According to the EC50 values, OPCs (EC50 = 2.93 µg/ml) and punicalagin (EC50 = 2.88 µg/ml) had higher potency against omicron variant virus production than tannic acid (EC50 = 5.27 µg/ml). Interestingly, the Vpp infection showed that punicalagin lost its inhibitory activity against viral entry of omicron variant Vpp (*Figure 4F*), but it still can efficiently repress the live omicron virus production. The potent inhibitory

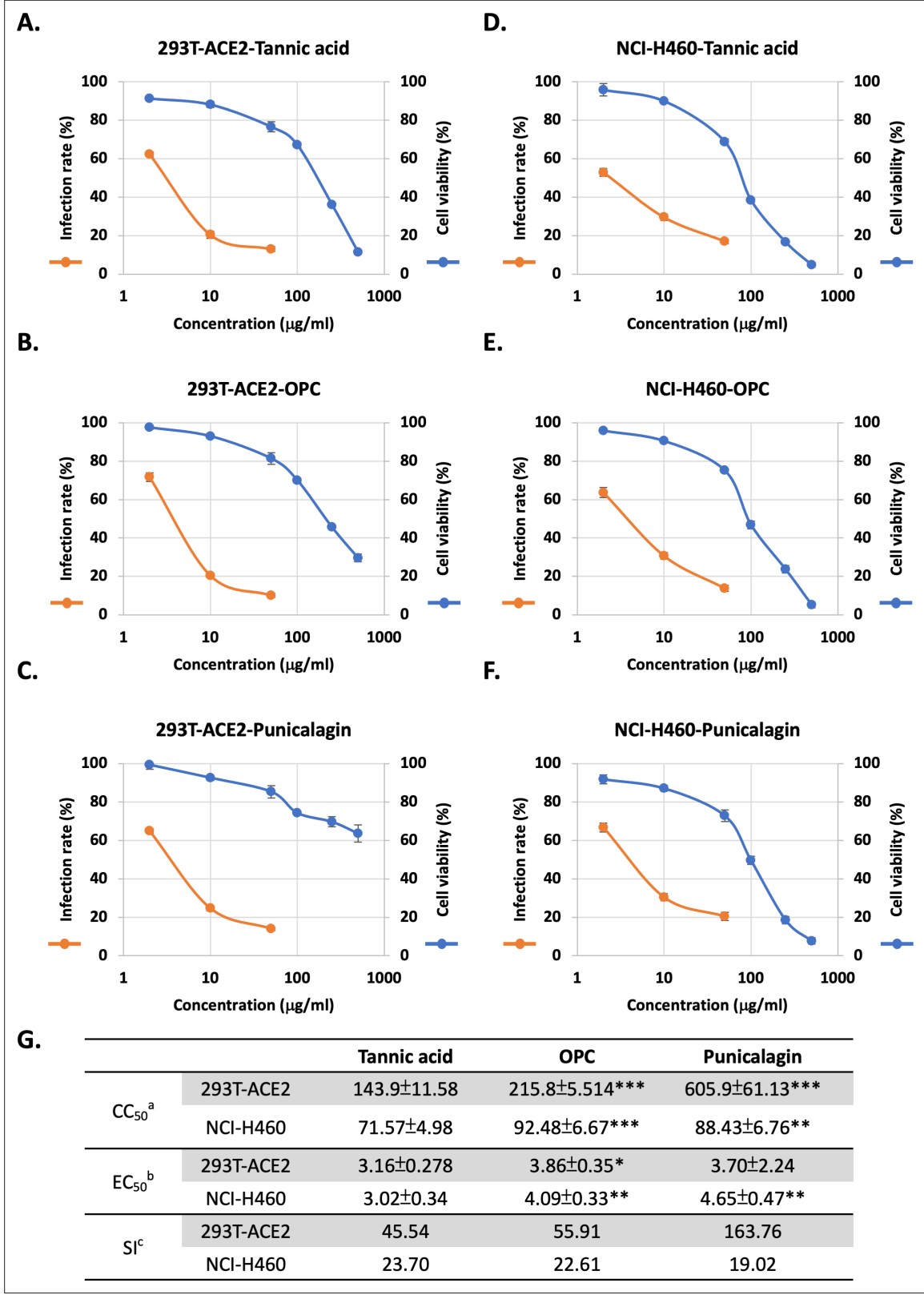

**Figure 2.** Three tannins could inhibit cell entry of Spike-viral pseudo-particles (Vpps) with low cytotoxicity in two cell lines. (**A–F**) Orange line: 293T-ACE2 and NCI-H460 cells were pretreated with varying concentrations (2, 10, and 50 µg/ml) of tannic acid, oligomeric proanthocyanidins (OPCs), or punicalagin for 1 hr and infected with wild-type SARS-CoV-2 Spike Vpps. After 24 hr of infection, the infection efficiency rate was measured according to luciferase activities. Blue line: 293T-ACE2 and NCI-H460 cells were treated with different concentrations (2, 10, 50, 100, 250, and 500 µg/ml) of tannic

*Figure 2 continued on next page*

*Figure 2 continued*

acid, OPCs, or punicalagin, and cell viability was detected with MTT assay. All data are shown as mean ± SD (n = 3). (**G**) Antiviral activity of tannic acid, OPCs, and punicalagin against SARS-CoV-2 Vpps based on (**A–F**). [a]CC50 is the median cytotoxic concentration such as the dose causing 50% cell death. [b]EC50 is the half-maximal effective concentration such as the concentration of a compound required to inhibit SARS-CoV-2 infection by 50%. [c]SI is the safety index such as the ratio of CC50 to EC50. For CC50 and EC50, extra-sum-of-squares $F$-test was performed to evaluate differences of OPC or punicalagion compared to tannic acid. The p-values are indicated by asterisks, *p<0.05; **p≤0.01; ***p≤0.001.

The online version of this article includes the following source data for figure 2:

**Source data 1.** The table dispays the 50% cytotoxic concentration (CC50), 50% effective concentration (EC50), and selectivity index (SI) from the experiments in *Figure 2A-F*.

activity of punicalagin on M[pro] that was required for virus replication but not for viral entry in the Vpp assay might contribute to the inhibition of omicron virus infection. These results suggested that all three tannin compounds could block SARS-CoV-2 omicron variant virus infection by inhibiting viral entry or replication via different mechanisms.

## OPCs-enriched GSE had the highest suppressive activity in inhibiting different variants of Vpps entry

Since all three tannin compounds associated with anti-SASR-CoV-2 activity through different mechanisms and they are enriched in fruits, this finding prompted us to examine the potential anti-omicron effect of tannin-enriched fruits. Grapes are a natural fruit containing abundant tannin (*Wang et al., 2020d*). The relative contents of OPCs, tannic acid, and punicalagin in different parts of the grapes produced from Changhua County in Taiwan were analyzed using ultra-high performance liquid chromatography coupled to high-resolution mass spectrometry (UHPLC-HRMS) (*Figure 6A–C*). Fractionation analysis showed that water extractants from the seeds contained much more OPCs compared to peel and flesh extractants (*Figure 6D*). However, each part of the grape extractants included a small amount of tannic acid and was almost free of punicalagin (*Figure 6D*). We further measure the concentration of OPCs in the juice from seedless grapes and regular grapes containing grape seeds. The result showed that seedless grapes contained far less level of OPCs than regular grapes with seeds (*Figure 6D*) and suggested that grape seeds were an important source of OPCs, the main compound of the three tannins. Then, we further examined the potential activity of water extractants from the seeds, peel, and flesh of the grapes using Vpp assay. The results showed that the seed extractant had the best inhibitory effect against SARS-CoV-2 wild-type Vpp infection (*Figure 6E*), consistent to the expectation from content of the enriched OPCs. Moreover, the extractants from different grape parts were tested in Vpp assays against alpha, beta, gamma, delta, and omicron variants (*Figure 6F*). In conclusion, GSE had the highest concentration of OPC and suppressive activity in inhibiting different variants of Vpps entry.

## Daily intake of GSE may be able to prevent SARS-CoV-2 infection

The results shown above suggested that GSE might have inhibitory activity against SARS-CoV-2 viral entry. To further address this issue, a human study was performed to investigate the effects of 200 mg or 400 mg daily intake of GSE on SARS-CoV-2 viral entry. As shown in *Figure 7A*, we purchased the GSE powder from raw materials suppliers and prepared the GSE capsules. Then, the healthy human subjects were randomly divided into three groups (0 mg, 200 mg, or 400 mg) to take GSE capsules or placebo and their serum samples were collected at the indicated time points (*Figure 7A*). The serum samples collected from the human subjects were further examined for their potential activity with Vpp assay. The results showed that serum samples from the human subjects who intake either 200 mg or 400 mg GSE capsules had a dose-dependent suppressive activity against different variants of SARS-CoV-2 Vpp infection compared to the baseline serum samples (*Figure 7B–F*). In addition, we also observed that serum samples from the human subjects who intook GSE capsules twice had better protection from wild-type, alpha, delta, and omicron SARS-CoV-2 Vpp infection than the serum samples derived from the subjects who intook GSE capsules only once. These results indicated that daily intake of GSE may be able to prevent SARS-CoV-2 infection.

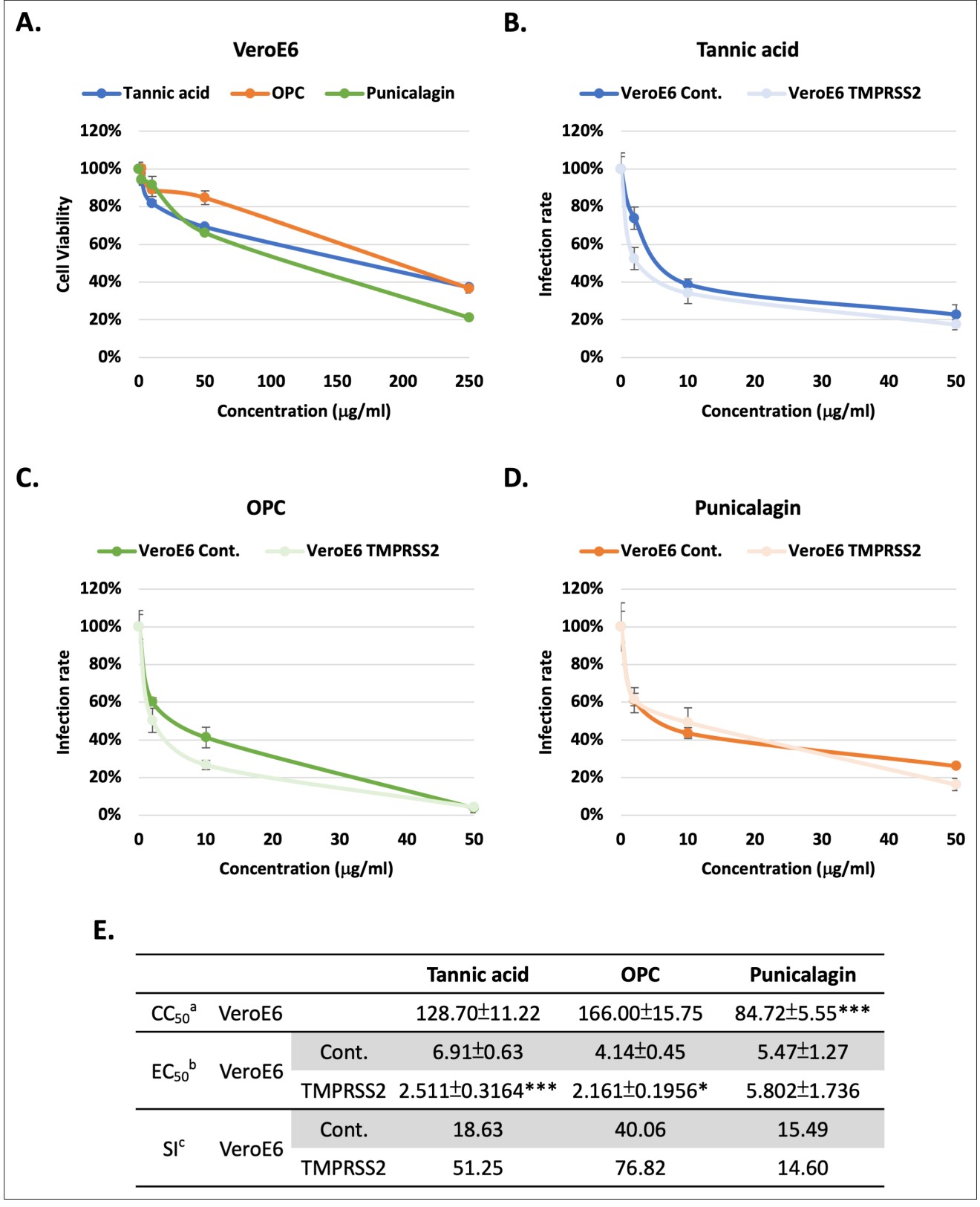

**Figure 3.** Tannic acid and oligomeric proanthocyanidins (OPC), but not punicalagin, had higher potency against viral pseudo-particles (Vpps) infection in the TMPRSS2-expressing cells. (**A**) VeroE6 cells were treated with indicated concentrations (2, 10, 50, and 250 µg/ml) of tannic acid, OPCs, or punicalagin, and cell viability was detected with MTT assay. (**B–D**) VeroE6 cells with and without TMPRSS2 expression were pretreated with the indicated concentrations (2, 10, and 50 µg/ml) of tannic acid, OPCs, or punicalagin, and then infected with SARS-CoV-2 wild-type Spike Vpps. After 24 hr of

*Figure 3 continued on next page*

*Figure 3 continued*

infection, the infection efficiency rate was measured according to luciferase activities. All data are shown as mean ± SD (n = 3). (**E**) Antiviral activity of tannic acid, OPCs, and punicalagin against SARS-CoV-2 Vpps based on (**A–D**). [a]$CC_{50}$ is the median cytotoxic concentration such as the dose causing 50% cell death. [b]$EC_{50}$ is the half-maximal effective concentration such as the concentration of a compound required to inhibit SARS-CoV-2 infection by 50%. [c]SI is the safety index such as the ratio of CC50 to EC50. For CC50, extra-sum-of-squares *F*-test was performed to evaluate differences of OPC or punicalagion compared to tannic acid. For EC50, extra-sum-of-squares *F*-test was performed to evaluate differences of tannic acid, OPC, or punicalagin of overexperssing TMPRSS2 cells compared to control cells. The p-values are indicated by asterisks, *p<0.05; **p≤0.01; ***p≤0.001.

The online version of this article includes the following source data for figure 3:

**Source data 1.** The table dispays the 50% cytotoxic concentration (CC50), 50% effective concentration (EC50), and selectivity index (SI) from the experiments in *Figure 3B-D*.

## Discussion

The current study demonstrates that the tannin derivatives are potent inhibitors of the SARS-CoV-2 virus and its variants. Given that the tannin compounds are enriched in natural products such as grape and pomegranate, tannin uptake from common fruits may have the benefit of reducing viral infection. We have reported that tannic acid is an inhibitor of the SARS-CoV-2 M[pro] with a binding pocket predicted by docking analysis involving the key residue Cys145 via the pi-sulfur and hydrogen bond (*Wang et al., 2020d*). Consistently, a recent in silico study showed interactions of tannins with the catalytic dyad residues (Cys-145 and His-41) of SARS-CoV-2 M[pro] (*Khalifa et al., 2020*). Similarly, structural docking analyses suggested that tannins have better binding affinities than remdesivir with the SARS-CoV-2 M[pro] protein (*Falade et al., 2021*). These studies together support the anti-COVID activities of the tannins.

In the current study, representative members of the tannin subfamilies, including OPCs of condensed tannin, punicalagin of ellagitannins, and the positive control tannic acid of gallotannins, were tested for anti-SARS-CoV-2 activities. Ellagitannins and gallotannins belong to hydrolysable tannins, usually made up of a glucose core, while the condensed tannins consist of oligomers or polymers of flavan-3-ol subunits (*Aboagye and Beauchemin, 2019*). Our results support the OPCs for its activity in SARS-CoV-2 inhibition. Generally, OPCs are comprised of the monomeric unit flavan-3-ols such as (+)catechin and (-) epicatechin and divided into propelargonidins, prodelphinidins, and procyanidins depending on the type of monomeric flavan-3-ols (*Bittner et al., 2013*). The most common type of proanthocyanidins are procyanidins, which are classified into monomer, dimer, trimer, and other polymers reflecting the degrees of polymerization (*Rue et al., 2018*). The OPCs with lower degrees of polymerization such as 2–4 monomers are more soluble and mobile, whereas the polymeric proan-thocyanidins consisting of >5 monomers are difficult to be absorbed in the body (*Unusan, 2020*; *Wang et al., 2020a*). Especially, dimeric and trimeric procyanidins have stronger antioxidant activity, better intestinal absorption, and better bioavailability in the gut than other types of procyanidins as demonstrated in in vitro studies, and animal as well as humans (*Deprez et al., 2001*; *García-Ramírez et al., 2006*). Besides, binding between M[pro] and the dimeric proanthocyanidins such as procyanidin B2 (PB2) has been modeled (*Zhu and Xie, 2020*). In vitro analysis showed that PB2 inhibited the M[pro] of SARS-CoV-2 at an $IC_{50}$ (half-maximal inhibitory concentration) of 75.3 ± 1.29 μM. Punicalagin is enriched in pomegranate and has been shown to have a unique inhibitory activity against the SARS-CoV-2 virus via blocking the Spike–ACE2 interaction and inhibiting the SARS-CoV-2 M[pro] (*Tito et al., 2021*). According to the inhibitory activities of the SARS-CoV-2 virus in the tannin family and our data performed in this paper, it is indicated that different classifications of tannin might be potential nutritional supplements to enhance protection against SARS-CoV-2 virus and its variants.

Multiple cell assay systems expressing infection-aiding proteins, including ACE2 and TMPRSS2 in the cell lines Vero E6, NCI-H460, and 293T, were employed in this report, coupled with the Vpps derived from the wild-type or the other variants. Assessments of the efficacy across these different systems unveiled potentially essential aspects of the action mechanisms of the tannins. In general, cells expressing TMPRSS2 were more sensitive to tannic acid and OPCs than the vector control, suggesting that the increased expression of TMPRSS2 conveyed a vulnerability to these two tannins. One possibility is that tannic acid and OPCs act at least in part through attenuating the virus-cell engagement. On the other hand, punicalagin did not have TMPRSS2 preference, suggesting a distinct mode of action compared to tannic acid and OPCs.

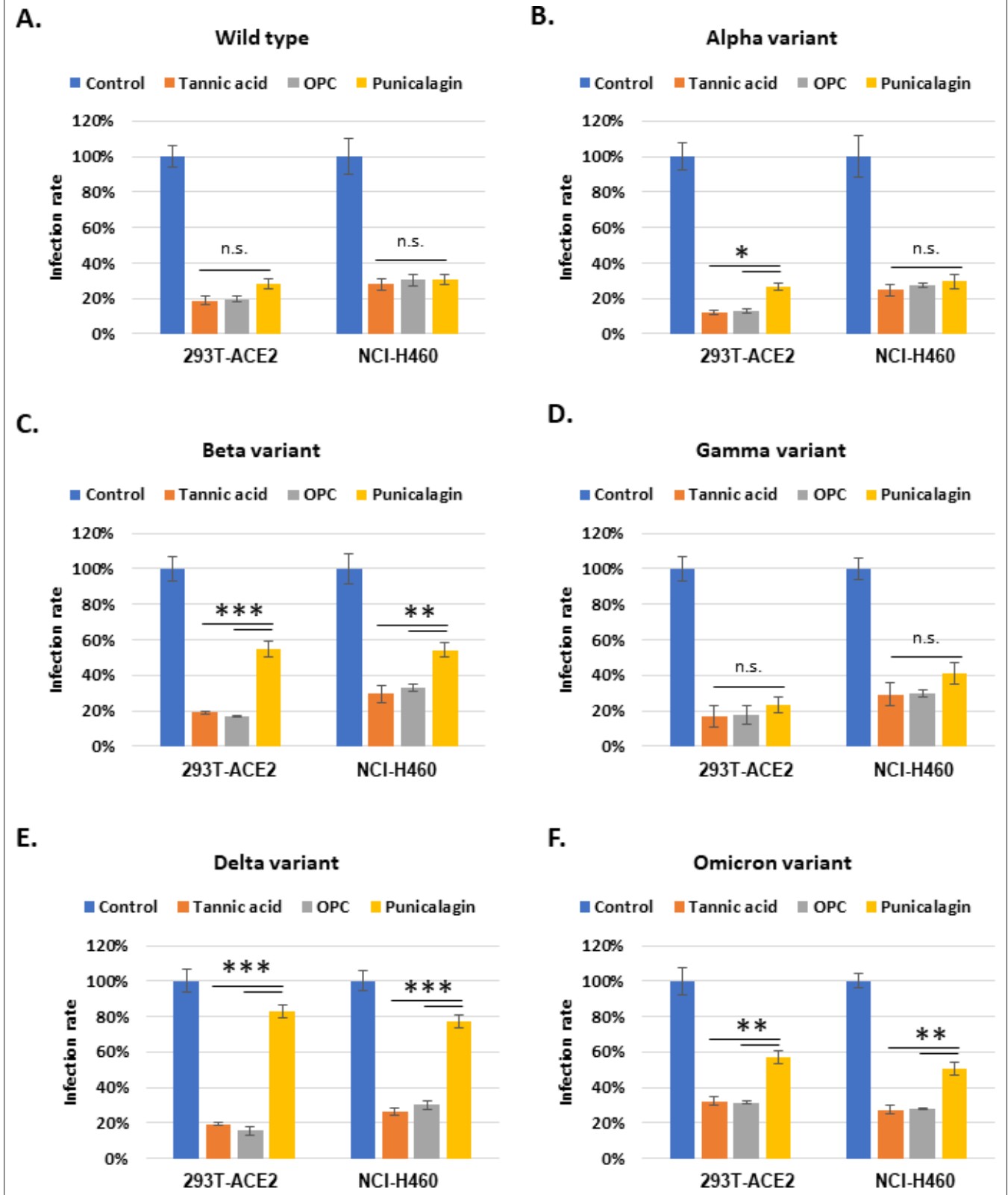

**Figure 4.** Tannic acid and oligomeric proanthocyanidins (OPCs), but not punicalagin, could maintain their inhibitory activity against different variants of viral pseudo-particles (Vpps) infection. (**A–F**) 293T-ACE2 and NCI-H460 cells were pretreated with 10 mg/ml tannic acid, OPCs, or punicalagin for 1 hr and infected with SARS-CoV-2 spike Vpps of different variants. After 24 hr of infection, the infection efficiency rate was measured according to luciferase activities. All data are shown as mean ± SD (n = 3). The p-values are indicated by asterisks, *p<0.05; **p≤0.01; ***p≤0.001; n.s., no statistical significance.

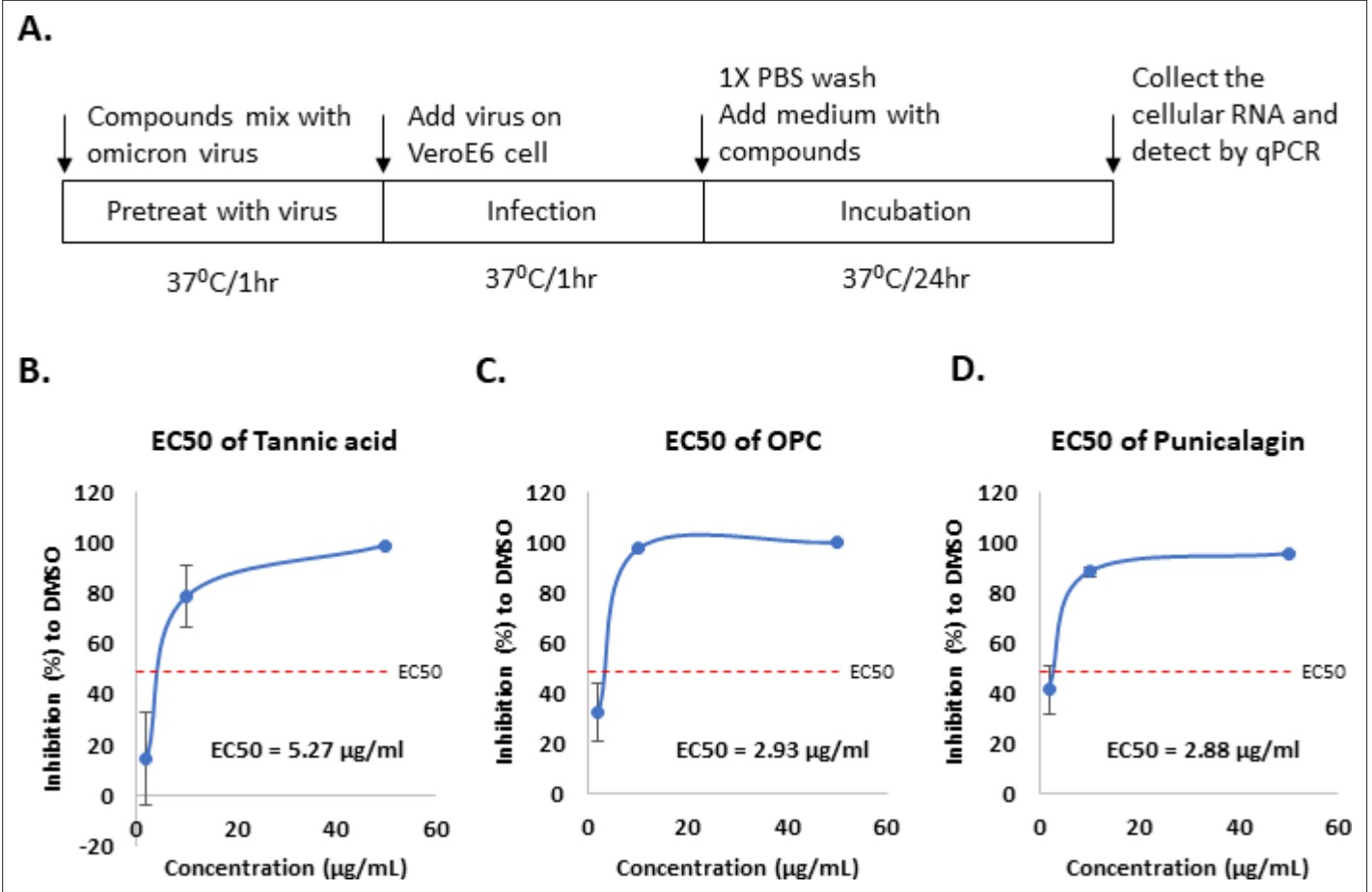

**Figure 5.** Oligomeric proanthocyanidins (OPC) and punicalagin could inhibit more effectively the infection of omicron virus compared to tannic acid. (**A**) The schematic illustrates tannic acid, OPCs, or punicalagin treatment of VeroE6 cells infected with the SARS-CoV-2 omicron virus. First, viruses were premixed with tannic acid, OPCs, or punicalagin (2, 10, and 50 µg/ml) for 1 hr before incubation with cells. Cells were continuously treated with tannic acid, OPCs, or punicalagin (2, 10, and 50 µg/ml) after infection with virus mixture for another 1 hr. (**B–D**) The inhibition rate of tannic acid, OPCs, or punicalagin against the SARS-CoV2 omicron virus was normalized with vehicle control. All data are shown as mean ± SD (n = 3).

Furthermore, the Vpp assays (*Figure 4*) in the 293T/ACE2 and NCI-H460 cells showed that tannic acid and OPCs could block alpha, beta, gamma, delta, and omicron variants Vpp infection as well as wild-type Vpps. Conversely, punicalagin is a potent inhibitor only for wild-type, alpha, and gamma Vpp infection, but not beta, delta, and omicron Vpps. Punicalagin prevented viral entry relying on block ACE2–Spike protein interaction. Therefore, different mutations of viral spike protein might make it prone to ineffectiveness. Otherwise, punicalagin had an excellent ability to obstruct virus M^pro enzyme activity (*Figure 1B*) and then might decline the virus replication. Consequently, punicalagin and OPCs were considered potent inhibitors against omicron variants in a live virus system (*Figure 5*). These results suggested that different tannin compounds may have overlapped but distinct action mechanisms in inhibiting viral infection.

While this article was under preparation, the omicron variant (B.1.1.529 or BA.1) had become the globally prevalent SARS-CoV-2 strain since late December 2021. BA.1 has since been replaced by emerging lineages BA.2 in March 2022, followed by BA.4 and BA.5, which have accounted for a majority of SARS-CoV-2 infections since late June 2022 (*Khan et al., 2022*; *US Department of Health and Human Services, 2023*). However, a recent report showed that vaccine effectiveness against hospitalization during the BA.1/BA.2 and BA.4/BA.5 periods was 79 and 60%, respectively, during the 120 d after the third dose and decreased to 41 and 29%, respectively, and indicated that the effectiveness of three doses of mRNA vaccines against COVID-19–associated hospitalization had declined (*Surie et al., 2022*). In addition, to measure neutralizing antibody titers after vaccination, the neutralization titer (NT50) was defined as the highest dilution at which the input virus signal was

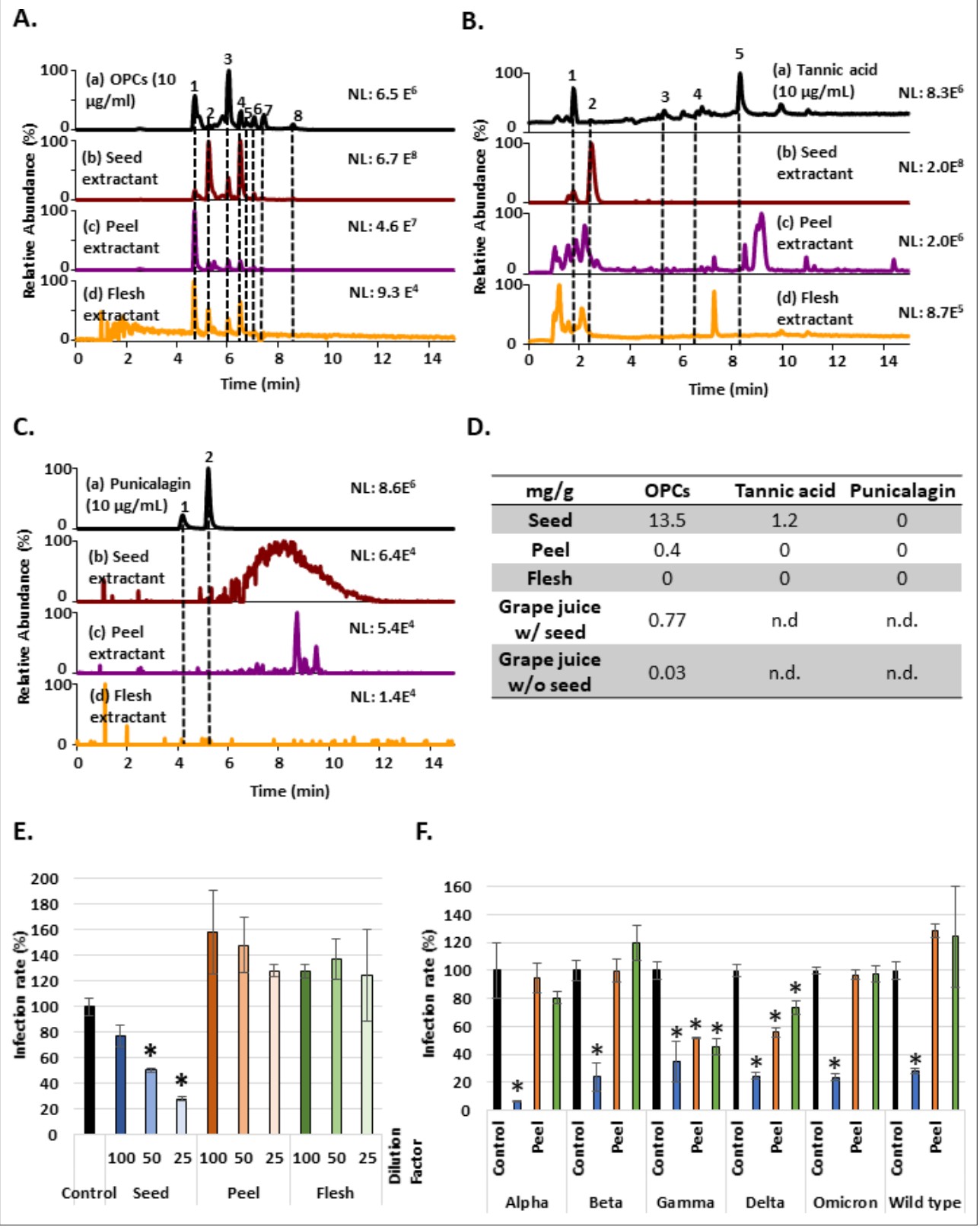

**Figure 6.** Grape seed extractant had the highest concentration of oligomeric proanthocyanidins (OPCs) and could inhibit the entry of wild-type and different variant Vpps. (**A**) Mass ion chromatograms of (a) OPCs (10 µg/ml), (b) grape seed extractant, (c) grape peel extractant, and (d) grape flesh extractant. Peaks 1 and 3: dimer of catechins; peaks 2 and 4: catechins; peaks 5 and 7: dimer of catechin-catechin gallate; peak 6: trimer of catechins; peak 8: catechin gallate. (**B**) Mass ion chromatograms of (a) tannic acid (10 µg/ml), (b) grape seed extractant, (c) grape peel extractant, and (d) grape

*Figure 6 continued on next page*

*Figure 6 continued*

flesh extractant. Peak 1: gallic acid; peak 2: galloylglucose; peak 3: digalloylglucose; peak 4: trigalloylglucose; peak 5: tetragalloylglucose. (**C**) Mass ion chromatograms of (a) punicalagin (10 µg/ml), (b) grape seed extractant, (c) grape peel extractant, and (d) grape flesh extractant. Peak 1: α-punicalagin; peak 2: β-punicalagin. (**D**) Based on the mass ion chromatograms results, OPC and tannic acid concentrations (mg/g) of different parts of grape water extractant or grape juice were converted from the concentration of the standard chemicals. n.d., non-determination. (**E**) 293T-ACE2 cells were pretreated with a different dilution factor of grape seed, peel, or flesh extractants for 1 hr and infected with wild-type SARS-CoV-2 spike Vpps. After 24 hr of infection, the infection efficiency rate was measured according to luciferase activities. (**F**) 293T-ACE2 cells were pretreated with grape seed, peel, or flesh extractants for 1 hr and infected with different variants of SARS-CoV-2 spike Vpps. After 24 hr of infection, the infection efficiency rate was measured according to luciferase activities. All data are shown as mean ± SD (n = 3). The p-values are indicated by asterisks compared to the control group (black column), *p<0.05.

The online version of this article includes the following source data for figure 6:

**Source data 1.** The table displays the OPC and Ttannic acid concentrations (mg/g) of different parts of grape water extractant or grape juice converted from the concentrations of the standard chemicals based on the mass ion chromatogram results.

reduced by at least 50%. Several studies indicated the mean of NT50 waned by 6 mo after vaccination and showed no neutralizing activity against omicron (*Edara et al., 2022*; *Favresse et al., 2023*; *Singanallur et al., 2022*). In conclusion, the results suggest that vaccination does not fully protect people from omicron virus infection after 4–6 mo of vaccination.

Thus, safe and effective food or nutritional supplements are urgently needed to protect people from this and other emerging mutant virus strains. Our results warrant further studies to elucidate the biological functions of tannins in infection inhibition and prevention. According to the viral load inhibition assay (*Figure 5D*), the EC50 of OPCs against omicron virus infection is 2.93 µg/ml. A typical adult human has approximately 5 l of blood in the body. Therefore, this EC50 converted to human blood content is 14.65 mg/5 l. Based on the mass spectrometry results (*Figure 5D*), the concentration of OPCs in GSE and grape juice with seed is 13.5 mg/g and 0.77 mg/g. Assuming that the absorption rate is 100%, only 1.085 g of the grape seeds or 19.025 g of grape juice can be consumed to achieve an effective concentration. Even though there is no clear cut-off to define an NT50 value that indicates protective immunity, most studies considered the NT50 < 100 was defined as a lack of neutralizing activity (*Edara et al., 2022*; *Kevlicius et al., 2022*). Therefore, we chose 200 times dilution of serum to evaluate the effectiveness of the GSE supplement. The data showed that 200 times dilution of serum from the human subjects who took GSE reduced by about 50% infection rate compared to baseline (*Figure 7*). It suggested that compared to vaccination alone, daily GSE supplement significantly increases prevention from SARS-CoV-2 infection. It is nonpractical to vaccinate people every 4–6 mo as it is not sustainable or affordable today. But we can power our body with GSE daily. Several studies have suggested that GSE had beneficial effects against many diseases, such as inflammation, cardiovascular disease, hypertension, diabetes, cancer, peptic ulcer, microbial infections, and so on (*Gupta et al., 2020*). Therefore, GSE was considered a healthy dietary supplement and was available as a liquid, tablet, or capsule. Previous studies have shown that the lethal dose (LD50) of GSE is >5000 mg/kg in the rat (*Lluís et al., 2011*). In another subchronic toxicity investigation with rats, the no-observed-adverse-effect level of GSE was 1410 mg/(kg body weight/day) for males and 1501 mg/(kg body weight/day) for females (*Yamakoshi et al., 2002*). Studies conducted in humans used in the range of 150–300 mg GSE daily for heart health purposes, while doses up to 600 mg have been used with no reported side effects (*Nowshehri et al., 2015*). Most importantly, the safety and tolerability of oral dosages of proanthocyanidin-rich GSE in healthy adult volunteers were evaluated in a clinical investigation. GSE up to 2500 mg taken orally for 4 wk was discovered to be typically safe and well tolerated in people (*Sano, 2017*). Our findings demonstrated that, in comparison to baseline blood samples, serum samples from human patients who took either 200 mg or 400 mg GSE capsules had a dose-dependent suppressive efficacy against various SARS-CoV-2 Vpp infection types. The daily consumption of 200 mg or 400 mg GSE is much lower than the safe dose shown in previous studies. The National Institute of Environmental Health Sciences conducted a toxicological assessment of GSE due to its potential health benefits to humans (*National Toxicology Program, 2023*). Following this evaluation, the Food and Drug Administration certified it as generally recognized as safe. Consequently, it is now marketed as a dietary additive and is listed on the Everything Added to Food in the United States database (*US Food and Drug Administration, 2022*). These results suggested that daily GSE supplements would be safe and provide benefits against COVID-19.

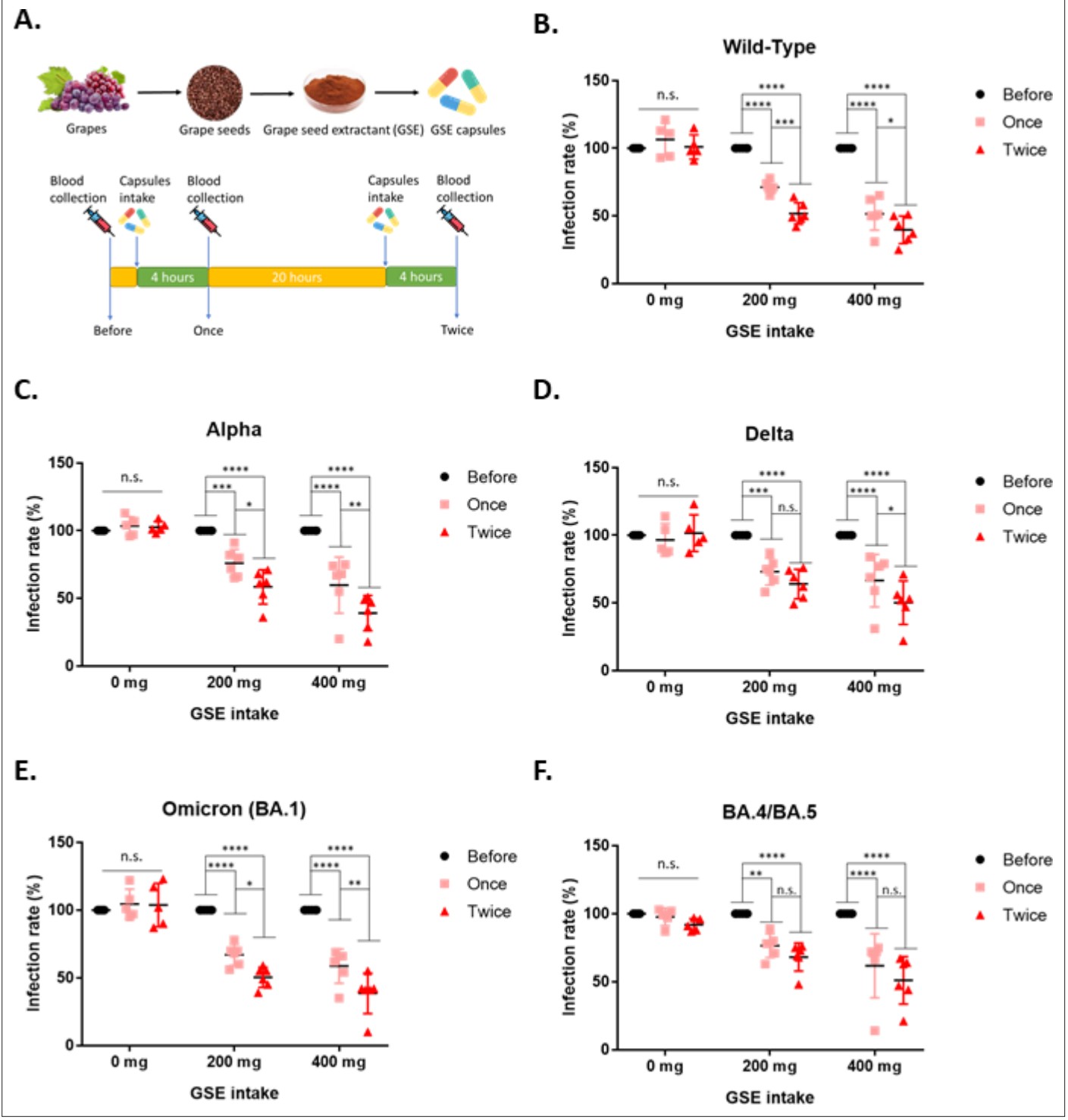

**Figure 7.** Serum from the human subjects who intake grape seed extractants had better potency to block the entry of wild-type and different variant Vpps. (**A**). The schematic illustrates the timeline of blood collection. First, the baseline blood sample was collected before taking control or grape seed extract (GSE) capsules. After baseline blood sample collection, the human subjects were randomized to one of three groups to take the GSE capsule (200 mg or 400 mg), or the placebo and collected the blood at the indicated time. (**B–F**) Each serum sample from the human subjects was diluted 1/200 and premixed with wild-type or different variants of SARS-CoV-2 spike Vpps for 1 hr before incubation with 293T-ACE2 cells. After 24 hr of infection, the infection efficiency rate was measured according to luciferase activities. All data are shown as mean ± SD (n = 5). The p-values are indicated by asterisks, *p<0.05; **p≤0.01; ***p≤0.001; ****p≤0.0001; n.s., no statistical significance.

## Conclusion

We found that two tannin compounds OPCs and punicalagin had potent inhibitory activity against SARS-CoV-2 virus infection, in addition to the tannic acid reported before. We further assessed the efficacy across the different cell systems and different variants of Vpps and unveiled potentially essential aspects of the action mechanisms of OPCs and punicalagin. The results indeed showed that OPCs and punicalagin can serve as good inhibitors against SARS-CoV-2 infection. Furthermore, we also observed that GSE contained abundant OPCs and may serve as a good resource for food supplements to prevent virus infection. It seems unlikely that SARS-CoV-2 will disappear naturally or by current intervention strategies (*Furuse and Oshitani, 2020*). Thus, identifying natural products such as fruits associated with anti-SARS-CoV-2 activities may be important for the community. The knowledge stemming from this study can help in the guideline for the development of food, nutrients, or supplements to prevent SARS-CoV-2 infection.

## Acknowledgements

We would like to acknowledge the service provided by the Biosafety Level-3 Laboratory of the First Core Laboratory, National Taiwan University College of Medicine. We thank the National RNAi Core Facility at Academia Sinica in Taiwan for providing all SARS-CoV-2 spike-pseudotyped lentiviruses and related services. This work was partially supported by grants from the National Science and Technology Council, Taiwan (NSTC 112-2639-B-039-001-ASP, MOST 110-2628-B-039-004 and MOST 109-2327-B-039-003) and Ministry of Health and Welfare Taiwan (MOHW112-TDU-B-222-124016). This work was also financially supported by the "Cancer Biology and Precision Therapeutics Center, China Medical University" from The Featured Areas Research Center Program within the framework of the Higher Education Sprout Project by the Ministry of Education (MOE) in Taiwan.

## Additional information

### Competing interests

Hsiao-Fan Chen, Wei-Jan Wang, Wei-Chao Chang, Yeh Chen, Shao-Chun Wang, Mien-Chie Hung: registered as the inventor of a patent application based on inhibiting SARS-CoV-2 infection of punicalagin and OPC (ROC Patent No.111133318). The other authors declare that no competing interests exist.

### Funding

| Funder | Grant reference number | Author |
|---|---|---|
| National Science and Technology Council | NSTC 112-2639-B-039-001 -ASP | Mien-Chie Hung |
| National Science and Technology Council | MOST 110-2628-B-039-004 | Shin-Lei Peng |
| Ministry of Health and Welfare, Taiwan | MOHW112-TDU-B-222-124016 | Mien-Chie Hung |
| Ministry of Education in Taiwan | "Cancer Biology and Precision Therapeutics Center, China Medical University", The Featured Areas Research Center Program, Higher Education Sprout Project | Mien-Chie Hung |
| National Science and Technology Council | MOST 109-2327-B-039-003 | Mien-Chie Hung |

The funders had no role in study design, data collection and interpretation, or the decision to submit the work for publication.

## Author contributions
Hsiao-Fan Chen, Data curation, Investigation, Methodology, Writing – original draft, Writing – review and editing; Wei-Jan Wang, Chen-Shiou Wu, Yeh Chen, Hsin-Yu Huang, Wan-Jou Shen, Investigation; Chung-Yu Chen, Investigation, Methodology; Wei-Chao Chang, Resources, Methodology; Po-Ren Hsueh, Resources, Investigation, Methodology; Shin-Lei Peng, Funding acquisition, Methodology; Shao-Chun Wang, Conceptualization, Supervision, Writing – original draft, Writing – review and editing; Mien-Chie Hung, Conceptualization, Supervision, Funding acquisition, Writing – original draft, Project administration

## Author ORCIDs
Mien-Chie Hung  https://orcid.org/0000-0003-4317-4740

## Ethics
Human subjects: The study was approved by the NTUH Research Ethics Committee (202101064RINB), and the participants gave written informed consent and consent to publish.

## Decision letter and Author response
Decision letter https://doi.org/10.7554/eLife.84899.sa1
Author response https://doi.org/10.7554/eLife.84899.sa2

# Additional files

## Supplementary files
• MDAR checklist

## Data availability
All data generated or analyzed during this study are included in the manuscript.

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
