## [Editor Report]

The study reports new findings that support the conclusion that natural tannins oligomeric proanthocyanidins and punicalagin are potent inhibitors of infection by SARS-CoV-2. The significant observations made by the authors could provide new strategies to prevent SARS-CoV-2 infection.

---

## [Decision Letter]

**Decision letter after peer review:**

Thank you for submitting your article "The natural tannins oligomeric proanthocyanidins and punicalagin are potent inhibitors of infection by SARS-CoV-2" for consideration by *eLife*. Your article has been reviewed by 2 peer reviewers, and the evaluation has been overseen by a Reviewing Editor and Wafik El-Deiry as the Senior Editor. The following individual involved in the review of your submission has agreed to reveal their identity: Chih-Chi Andrew Hu (Reviewer #1).

Essential revisions:

The manuscript needs improvement by addressing several issues as outlined below:

Reviewer #1 requested corrections and clarifications. This reviewer also agreed with Reviewer #2 after seeing the comments made by Reviewer #2. Regarding the three comments made by Reviewer #2, please address all three of them, including the concern about compound safety.

*Reviewer #1 (Recommendations for the authors):*

I have only a few suggestions that may help the authors improve their manuscript.

1. Line 302, should be corrected to "tannic acid, OPCs, and punicalagin".

2. Figures 3A and 3E were not called and discussed in Results.

3. In Figure 6D, please indicate the unit used. Is this from a single grape berry?

*Reviewer #2 (Recommendations for the authors):*

For all figures, statistical analyses should be done for all panels. As it stands, only some of the figure panels contain statistical notations.

1)For all figures, statistical analyses should be performed for all included panels. As it is currently presented, only select panels have statistical analyses indicated.

2) Additionally, for all IC50 or EC50 data, comparisons should be performed using appropriate statistical tests, namely Extra-sum-of-squares F-test to compare EC50 and IC50 curves. The statistical tests used in the current draft are not explicitly stated, however selective comparison of individual data points is not the standard for comparison of these types of data.

3) The authors may consider assessing the safety profiles of the proposed compounds using in vivo models and assess liver toxicity or cardiotoxicity using commercially available tests. This will also determine the doses at which GSE should be consumed to reach efficacious doses in human patients.

---

## [Author Response]

Reviewer #1 (Recommendations for the authors):I have only a few suggestions that may help the authors improve their manuscript.1. Line 302, should be corrected to "tannic acid, OPCs, and punicalagin".

Thanks. We corrected the description in Line 312-314.

2. Figures 3A and 3E were not called and discussed in Results.

Thanks for your comments. We have added descriptions for Figure 3A and 3E in “Result 3.2” (Line 320-327).

3. In Figure 6D, please indicate the unit used. Is this from a single grape berry?

Thanks. The unit of concentration in Figure 6D is mg/g which has been noted in the figure and figure legend of Figure 6D. Based on the mass ion chromatograms results, OPC and Tannic acid concentrations (mg/g) of different parts of grape water extractant or grape juice were converted from the concentration of the standard chemicals.

Reviewer #2 (Recommendations for the authors):For all figures, statistical analyses should be done for all panels. As it stands, only some of the figure panels contain statistical notations.1)For all figures, statistical analyses should be performed for all included panels. As it is currently presented, only select panels have statistical analyses indicated.

Thanks. We have checked all panels again to make sure statistical analyses were appropriately performed and described in the figure legends. We also have added descriptions for statistic analysis in “Method 2.13” (Line 266-269).

2) Additionally, for all IC50 or EC50 data, comparisons should be performed using appropriate statistical tests, namely Extra-sum-of-squares F-test to compare EC50 and IC50 curves. The statistical tests used in the current draft are not explicitly stated, however selective comparison of individual data points is not the standard for comparison of these types of data.

Thanks for the comments. The reviewer is right. We performed the extra-sum-of-squares F-test to compare the CC50 or EC50 curve of tannic acid, OPC, or punicalagin and added a description in the figure legends.

3) The authors may consider assessing the safety profiles of the proposed compounds using in vivo models and assess liver toxicity or cardiotoxicity using commercially available tests. This will also determine the doses at which GSE should be consumed to reach efficacious doses in human patients.

Thanks for the valuable comments. Regarding the concern, we agree with the reviewer the safety of GSE is indeed important. As a matter of facts, the safety issue of GSE has been documented in the literature. We apologize that we did not discuss this in detail in the original manuscript. We have now added a paragraph in the Discussion section to summarize published reports on the safety of GSE in vivo (Line 515-536).

Briefly, previous studies have shown that the lethal dose (LD50) of GSE is greater than 5000 mg/kg in the rat [1]. In another sub-chronic toxicity investigation with rats, the no-observed-adverse-effect level (NOAEL) of GSE was 1410 mg/(kg body weight/day) for males and 1501 mg/(kg body weight/day) for females [2]. Studies conducted in humans used in the range of 150-300mg GSE daily for heart health purposes, while doses up to 600mg have been used with no reported side effects [3]. Most importantly, the safety and tolerability of oral dosages of proanthocyanidin-rich grape seed extract (GSE) in healthy adult volunteers were evaluated in a clinical investigation. GSE up to 2500 mg taken orally for 4 weeks was discovered to be typically safe and well tolerated in people [4]. Our findings demonstrated that, in comparison to baseline blood samples, serum samples from human patients who took either 200 mg or 400 mg GSE capsules had a dose-dependent suppressive efficacy against various SARS-CoV-2 Vpp infection types. The daily consumption of 200mg or 400mg GSE is much lower than the safe dose shown in previous studies. The National Institute of Environmental Health Sciences conducted a toxicological assessment of grape seed extractant due to its potential health benefits to humans [5]. Following this evaluation, the Food and Drug Administration certified it as generally recognized as safe. Consequently, it is now marketed as a dietary additive and is listed on the Everything Added to Food in the United States database [6]. These results suggested that daily GSE supplements would be safe and provide benefits against COVID-19.

References

1. Lluis, L., et al., *Toxicology evaluation of a procyanidin-rich extract from grape skins and seeds.* Food Chem Toxicol, 2011. 49(6): p. 1450-4.

2. Yamakoshi, J., et al., *Safety evaluation of proanthocyanidin-rich extract from grape seeds.* Food Chem Toxicol, 2002. 40(5): p. 599-607.

3. Nowshehri, J.A., Z.A. Bhat, and M.Y. Shah, *Blessings in disguise: Bio-functional benefits of grape seed extracts.* Food Research International, 2015. 77: p. 333-348.

4. Sano, A., *Safety assessment of 4-week oral intake of proanthocyanidin-rich grape seed extract in healthy subjects.* Food Chem Toxicol, 2017. 108(Pt B): p. 519-523.

5. National Toxicology Program, N.I.o.E.H.S. *Testing Status of Grape seed extract M020012*. Available from: https://ntp.niehs.nih.gov/static/whatwestudy/testpgm/status/ts-m020012.html#GeneticToxicology.

6. Administration, U.S.F.a.D. *Substances Added to Food (formerly EAFUS)*. Available from: https://www.cfsanappsexternal.fda.gov/scripts/fdcc/?set=FoodSubstances.